# Suboptimal Plasma Vitamin C Is Associated with Lower Bone Mineral Density in Young and Early Middle-Aged Men: A Retrospective Cross-Sectional Study

**DOI:** 10.3390/nu14173556

**Published:** 2022-08-29

**Authors:** Kuo-Mao Lan, Li-Kai Wang, Yao-Tsung Lin, Kuo-Chuan Hung, Li-Ching Wu, Chung-Han Ho, Chia-Yu Chang, Jen-Yin Chen

**Affiliations:** 1Department of Anesthesiology, Chi Mei Medical Center, Liouying, Tainan 73657, Taiwan; 2Department of Anesthesiology, Chi Mei Medical Center, Tainan 71004, Taiwan; 3Department of Hospital and Health Care Administration, College of Recreation and Health Management, Chia Nan University of Pharmacy and Science, Tainan 71710, Taiwan; 4Center for Precision Medicine, Chi Mei Medical Center, Tainan 71004, Taiwan; 5Institute of Biomedical Sciences, National Sun Yat-Sen University, Kaohsiung 80424, Taiwan; 6Department of Medical Research, Chi Mei Medical Center, Tainan 71004, Taiwan; 7Department of Neurology, Chi Mei Medical Center, Tainan 71004, Taiwan; 8The Center for General Education, Southern Taiwan University of Science and Technology, Tainan 71004, Taiwan

**Keywords:** suboptimal plasma vitamin C, bone mineral density, lumbar spine, young adults, early middle-aged

## Abstract

Background: This study was conducted to evaluate associations between bone mineral density (BMD) and four selected circulating nutrients, particularly vitamin C, among adults aged 20–49 years. Methods: In this retrospective cross-sectional study, the lumbar spine BMD of 866 men and 589 women were measured by dual-energy X-ray absorptiometry and divided into tertiles, respectively. Logistic regressions were used to identify the predictors of low BMD by comparing subjects with the highest BMD to those with the lowest. Results: Multivariate logistic regressions identified suboptimal plasma vitamin C (adjusted odds ratio (AOR) 1.64, 95% confidence interval (CI) 1.16, 2.31), suboptimal serum vitamin B12 (AOR 2.05, 95% CI 1.02, 4.12), and low BMI (BMI < 23) (AOR 1.68, 95% CI 1.12, 2.53) as independent predictors for low BMD in men. In women, low BMI was the only independent predictor for low BMD. Plasma vitamin C, categorized as suboptimal (≤8.8 mg/L) and sufficient (>8.8 mg/L), was positively significantly correlated with the lumbar spine BMD in men, but there was no association in women. Conclusions: Plasma vitamin C, categorized as suboptimal and sufficient, was positively associated with the lumbar spine BMD in young and early middle-aged men. A well-designed cohort study is needed to confirm the findings.

## 1. Introduction

Osteoporosis is a progressive skeletal disease characterized by decreased bone mineral density (BMD) and microarchitectural deterioration of the skeleton, leading to bone fragility and a predisposition to fractures. Osteoporosis and the resulting fragility fractures are a growing public health problem. Osteoporosis has no clinical manifestations until there is a fracture. These fractures are responsible for lasting disability and impaired quality of life, with an enormous healthcare burden on both the patient’s and the nation’s economy [1,2,3]. In particular, the mortality risk following osteoporotic fractures significantly increases for 5 to 10 years [1]. However, osteoporosis is a preventable disease. It is imperative to identify risk factors that can be modified to prevent osteoporosis [3,4,5,6].

Peak bone mass is the maximum amount of bone during the life of an individual. Approximately 60–80% of peak bone mass is determined by genetic factors, and the remaining 20–40% is contributed by environmental factors, including lifestyle, physical activity, and nutrition [5]. Nutritional factors have been identified to have a role in the incidence of osteoporosis and bone fractures [3,4,5,6]. Vitamin C, also known as ascorbic acid, is an essential micronutrient for humans. Despite its enormous involvement in many physiological and biological functions, humans are unable to synthesize vitamin C due to mutations in the L-gulono-gamma-lactone oxidase gene, which codes for the enzyme in the last step of vitamin C biosynthesis [7]. Vitamin C is the essential cofactor for the two enzymes required for collagen synthesis: prolyl hydroxylase and lysyl hydroxylase. In the animal models of osteoporosis, vitamin C was observed to increase type I collagen (the main component of the organic part of bone) and stimulate the expression of osteocalcin (a bone formation marker) and osteogenesis-related proteins [8,9]. Overall, vitamin C is involved in improving bone regeneration and inhibiting osteoporosis via activating osteoblastogenesis and inhibiting osteoclastogenesis [8,9,10]. A recent study from US adults aged 46 to 78 years, with 72% women, reported that either dietary vitamin C or use of vitamin C supplements was not associated with BMD [4]. However, a systemic review and meta-analysis in 2018 revealed that greater dietary vitamin C intake was associated with a lower risk of osteoporosis and a higher BMD at the lumbar spine and femoral neck [11]. Evidence in the field of associations between vitamin C intake and BMD is inconsistent across studies. Vitamin C intake is a key determinant of body status; however, the correlation between dietary and circulating vitamin C can be affected by many factors, including body mass index (BMI) [12], smoking [13], alcohol consumption [14], vitamin bioavailability, absorption condition, food processing, and chronic diseases [15,16]. Conflicting evidence for the relation between vitamin C and bone health may be due to the lack of assessment of circulating vitamin C status. Therefore, it is important to assess the association between circulating vitamin C and BMD, which may better reflect the actual relationship between vitamin C and bone health [4,17,18]. Thus far, few studies [4,17,18] assess the potential relationship between plasma vitamin C and bone health, particularly in young and early middle-aged adults.

Deficiency of folic acid, vitamin B12, and vitamin D has been linked to the disturbance of bone metabolism. In cell models, vitamin B12 deficiency has no effect on osteoblast differentiation but causes the increased secretion of homocysteine by osteoblasts. Consequently, elevated homocysteine stimulates osteoclastogenesis in a dose-dependent manner [19]. Similarly, lower concentrations of either folic acid alone or the combination of folic acid and vitamin B12 increase osteoclast activity 2–3-fold compared to that at the initial physiologic concentration [20]. Overall, low folic acid and/or vitamin B12 can increase bone absorption via the enhancement of osteoclast activity in cells [19,20]. However, the findings of observational studies are inconsistent [18,21]. Vitamin D, a secosteriod hormone, is essential for calcium absorption and bone mineralization, which has a positive association with BMD. Although severe vitamin D deficiency leading to rickets in infants and children is well-established, the association of BMD with vitamin D in young adults was inconclusive [22,23]. The synergistic interplay of multiple vitamin deficiencies for bone has been shown [24,25]. Therefore, it is worthy to examine the associations between BMD and multiple circulating nutrients’ status, i.e., vitamin C, vitamin D, folic acid, and vitamin B12.

The concerted and coordinated activity of osteoclasts (bone-resorbing cells) and osteoblasts (bone-forming cells) constantly remodels bone. Bone is a dynamic tissue. In the whole body, the spine is more metabolically active than the hip or forearm; therefore, a significant change in BMD is likely to occur earlier at the spine [26]. BMD is the amount of bone mass per unit volume (volumetric density, g/cm^3^), or per unit area (areal density, g/cm^2^) [27]. BMD measured by dual-energy X-ray absorptiometry provides a two-dimensional areal value [27]. DXA is an internationally accepted standard-of-care screening tool for osteoporosis assessment. A low BMD is associated with an increased fracture risk. In this retrospective cross-sectional study, the lumbar spines BMD of men and women were measured by DXA and divided into tertiles (three equal parts), respectively. Low BMD was defined as BMD in the lowest tertile of the lumbar spine BMD [28]. In this study, we examined associations between the lumbar spine BMD and the four selected circulating nutrients (vitamin C, vitamin D, vitamin B12, and folic acid) as well as anthropometric characteristics (body weight, body mass index (BMI)), all of which have been shown to be interrelated with BMD and the occurrence of osteoporosis [17,29,30]. Primary outcomes of the study were to determine possible associations between selected circulating nutrients and low BMD at the lumbar spine by comparing subjects with the highest BMD to those with the lowest BMD among young and early middle-aged men and women, respectively. The secondary outcomes were to explore the distribution difference of the BMD at the spine among three categories of plasma vitamin C (i.e., deficient, insufficient, and sufficient) in men and women, respectively. The secondary outcomes were to explore the distribution difference of the BMD at the spine between suboptimal and sufficient plasma vitamin C in men and women, respectively.

## 2. Materials and Methods

This retrospective cross-sectional study was approved by the Institutional Review Board of the Chi Mei Medical Center, a 1200-bed tertiary referral hospital in Tainan, Taiwan [31,32,33,34]. This study was performed in accordance with the Declaration of Helsinki.

### 2.1. Study Population

The Taiwan government provides a universal, mandatory national health insurance, with a coverage rate up to 99% of Taiwan’s entire population by the end of 2004 [35]. The periodic health examination in adults is an insured service. Chi Mei Medical Center provides options on circulating nutrition surveys, including vitamin C, vitamin D, vitamin B12, and folic acid [36,37], which are not insured and served as a bundled package. If adults receiving periodic health examinations chose an option of nutrition surveys, the considered vitamins were all measured. Data of participants having health examinations at our hospital were routinely stored in the electronic database of the institute. Data of health examinations at Chi Mei Medical Center between 1 January 2019 and 31 December 2020 were retrospectively collected and analyzed. Inclusion criteria for this study were: (1) adults aged 20 to 49 years [18], (2) participants who had the DXA examination, and (3) those with available data of plasma vitamin C, serum 25-hydoxyvitamin D [25(OH)D], vitamin B12, and folic acid during the health examination period. 

Exclusion criteria were: (1) Adults with diabetes or prediabetes (International Classification of Diseases, ICD) (ICD-10 E10–E14, R73.09, or HbA1C ≥ 5.7% [38,39], or fasting glucose ≥ 120 mg/dL [4]), which may be a confounder in the relationship of BMD and vitamin C status because patients with type 1 diabetes often have a lower BMD. Patients with type 2 diabetes or prediabetes have a normal or even higher BMD as compared with healthy subjects [40]. (2) Menopausal women who have had no menstrual period for 12 months were excluded due to rapid bone loss after the menopausal interval [17,41]. (3) Subjects who had diagnostic codes of human immunodeficiency virus infection (ICD-10 B20–B24) [42], solid organ transplantation (ICD-10 Z94) [43], chronic liver diseases (ICD-10 K70–77) [44] and/or chronic kidney disease (ICD-10 N18.5, N18.6, or the estimated glomerular filtration rate (eGFR) < 60 mL/min/1.73 m^2^) [45], hyperthyroidism (ICD-10 E05) [46], hyperparathyroidism (ICD-10 E21) [47], Cushing’s syndrome (ICD-10 E24) [48], hypogonadism (ICD-10 E29.1) [49], and rheumatological diseases (ICD-10 M05, M32–35) [50], which are potential confounding factors in the relationship of BMD and vitamin status. (4) Participants who took medications (i.e., corticosteroids (Anatomical Therapeutic Chemical classification system (ATC)-D07) [51], some antiepileptic (ATC-N03) [52], and psychiatric drugs (ATC-05A) [53]) were excluded because these medications affect bone metabolism. (5) Smokers—individuals’ smoking behavior may have biased the interpretation of the study results [54]. (6) Adults whose medical records showed no evidence of all the targeted health examinations during the study period.

### 2.2. Study Parameters, Definitions, and Cut-Offs

Adult subjects aged 20–49 years were enrolled and divided into two age groups: young (20–35) and early middle-aged (36–49) [55]. Body mass index (BMI) was calculated as the weight in kilograms divided by the square of height in meters (kg/m^2^). Based on the WHO Asian BMI categories, all subjects in this study were graded and divided into four groups: underweight (BMI < 18.5), normal weight (BMI 18.5–22.9), overweight (BMI 23–27.4), and obesity (BMI ≥ 27.5) [56]. Individuals with underweight and normal weight are considered as having a low BMI (BMI < 23), whereas subjects with overweight and obesity are considered as having a high BMI (BMI ≥ 23).

In current international and national recommendations, a fasting plasma vitamin C concentration greater than (>8.8 mg/L, ≥50 µmol/L) is considered as sufficient. Suboptimal plasma vitamin C (≤8.8 mg/L, <50 µmol/L) [57] includes insufficient (6.1–8.8 mg/L, 34.3–49 µmol/L) [58] and deficient levels (≤6.0 mg/L, 34.2 μmol/L) [59]. For serum vitamin B12 status, the two cut-off points of <200 pg/mL (deficiency) and 200 to 300 pg/mL (insufficiency, marginal deficiency) were conventionally accepted cut-off points [60]. Suboptimal serum vitamin B12 status includes vitamin B12 insufficiency and deficiency [61]. Serum folic acid levels are considered sufficient at ≥6 ng/mL, insufficient at 3–5.9 ng/mL, and deficient at <3 ng/mL [62]. Suboptimal serum folic acid status includes folic acid insufficiency and deficiency. While these cut-off points may be considered somewhat arbitrary, there is a pathophysiological rationale for their use.

Serum 25-hydroxyvitamin D (25(OH)D) is often used as a marker of vitamin D status [63,64]. Based on the 2011 Endocrine Society Clinical Practice Guidelines, vitamin D status is defined as sufficient (optimal) (i.e., serum 25(OH)D concentration: 30–150 ng/mL) and suboptimal serum 25(OH)D, which includes insufficiency (20–30 ng/mL) and deficiency (<20 ng/mL) [34,63,65].

### 2.3. Assessment of Bone Mineral Density and Plasma/Serum Nutrient Status

#### 2.3.1. Determination of Bone Mineral Density and the Rate of Osteoporotic Fracture

On the health examination day, all subjects changed into gowns and were asked to remove all jewelry and other personal effects that could interfere with the DXA exam (Hologic, Inc., Bedford, MA, USA) [27]. DXA systems are available as either full-table systems or as peripheral systems (limited to measuring the peripheral skeleton). A full-table DXA system can measure BMD (g/cm^2^) at any skeletal site. Clinical use has been concentrated on the lumbar spine because the lumbar spine serves as the principal site for osteoporosis diagnosis and therapeutic decision-making [27]. Therefore, the lumbar spine scans using DXA are conducted as an option of the health examination. Osteoporotic fracture (ICD-10 Z87.310) is a common complication of osteoporosis [66]. The rate of osteoporotic fracture was investigated.

#### 2.3.2. Blood Collection and Determination of Plasma Vitamin C Concentrations

The fasting blood sample of every participant was drawn in the morning of the day when the participant received the health examination. The plasma sample after centrifugation was stored at a temperature of −80 °C until analysis. The plasma vitamin C quantification was regularly performed once a week by ultra-performance liquid chromatography (A chromatograph Acquity UPLC Waters*^®^* [67]), as in our previous report [37].

#### 2.3.3. Blood Collection and Determination of Serum 25(OH)D Concentrations

Serum levels of 25(OH)D were measured by automated immunoassay with ARCHITECTi2000SR (Abbott, Chicago, IL, USA) (Chemiluminescent Microparticle Immuno-Assay) [68]. A fasting blood sample of each subject was drawn and stored at 2–8 °C after centrifugation. The measurement of 25(OH)D was completed within 4 h after blood sampling, as in the previous description [34,37].

#### 2.3.4. Blood Collection and Determination of Serum Vitamin B12 and Folic Acid Concentrations

A fasting blood specimen was drawn from the subjects in the morning. Blood samples were collected in serum separator tubes with light protection and kept frozen at a temperature of −70 °C until analysis due to the unstable properties of vitamin B12 and folic acid [69]. The ARCHITECT vitamin B12/folate calibrators were used for the calibration of the ARCHITECT i system when used for the quantitative determination of serum vitamin B12/folate. Serum vitamin B12 and folic acid concentrations were measured with the ARCHITECT vitamin B12 and Folate Reagent Kit (Chemiluminescent Microparticle Immuno-Assay). The quantification of serum vitamin B12 and folic acid was performed every weekday.

### 2.4. Statistical Analysis

All analyses were performed using Statistical Analysis System (SAS) statistical software (version 9.4; SAS Institute, Inc., Cary, NC, USA). The Kolmogorov–Smirnov test was used to test for normality of the data. Comparison of the non-normally distributed data was performed using the Mann–Whitney U-test. The chi-square test was used to determine the significance of differences in categorical variables.

Bone mineral density measured by DXA is a direct method and a good indicator of bone health. A T-score is a standard deviation showing the difference between the BMD of a subject and the average of healthy 30-year-old adults. BMD is often used, not T-score, in the studies including youth [70]. A Z-score compares the bone density to the average values for a person of the same age and gender. A low Z-score (below −2.0) is a warning sign for less bone mass than expected for the age. There were only 8% of men and 2% of women to have a Z-score below −2.0 in our study population. Therefore, BMD was tertiled (three equal groups) to identify a susceptible population in men and women, respectively [71,72].

The primary outcomes were to identify possible associations between selected circulating nutrients and low BMD at the spine among young and early middle-aged men and women, respectively. Univariate logistic regression analysis was used to identify the predictors for low BMD by comparing demographic and anthropometric characteristics as well as laboratory results in subjects with the highest spine BMD to those with the lowest spine BMD. To identify the independent predictors for low BMD by multiple logistic regressions, confounding factors including age, BMI, vitamin C, 25(OH)D, vitamin B12, folic acid, and season were adjusted to estimate adjusted odds ratios (OR) and their 95% CI in multiple logistic models. A two-sided *p*-value < 0.05 was considered significant. In addition, the distribution differences between the BMD at the spine and the two categories of plasma vitamin C (i.e., suboptimal (≤8.8 mg/L) and sufficient (>8.8 mg/L)) in males and females were conducted using the Mann–Whitney U-test if non-normally distributed after Kolmogorov–Smirnov testing.

## 3. Results

### 3.1. Demographic/Anthropometric Characteristics and Laboratory and Radiology Results of the Study Population

Electronic records of 1455 subjects were included after application of the inclusion and exclusion criteria. In Table 1, laboratory and radiology results are shown.

The majority of the study population was men (866 cases, 59.5%). High BMI (overweight and obesity) was more common in men (77.5%) than women (31.6%) (*p* < 0.001). Participants did not differ by age (*p* = 0.097) and alcohol consumption (*p* = 0.983), whereas the sample had a significant difference in height, weight, BMI, and the four selected circulating nutrients (all *p* < 0.001). As for BMD, men obtained lower bone mineral density values at the spine than women (*p* < 0.001). No osteoporotic fracture was found in both genders of the study population.

The prevalence of suboptimal plasma vitamin C, serum vitamin B12, and serum folic acid was more prominent in men (*p* < 0.001, *p* = 0.002, *p* < 0.001). Women were more likely to have suboptimal serum 25(OH)D status (insufficient and deficient serum 25(OH)D) than men (81.7% vs. 66.5%) (*p* < 0.001).

### 3.2. Distribution of Suboptimal Circulating Nutrients in the Four Selected Nutrients

Only 117 men (13.5%) and 75 women (12.7%) did not have suboptimal circulating nutrients for any of the 4 selected nutrients. The overall prevalence of suboptimal nutrients was up to 86.8% (86.5% in men and 87.3% in women). The average count of suboptimal nutrients for the four nutrients in men was 1.5 (SD 1.0), which was significantly higher than that (1.2, SD 0.7) in women. Considering age, 93.8% of young subjects had suboptimal nutrients for any of the four nutrients; meanwhile, the proportion of middle-aged subjects with suboptimal nutrients for any of the four nutrients was 85.9% (*p* < 0.001). As for the lumbar spine BMD, the average count (1.5, SD 0.9) of suboptimal nutrients in the lowest BMD tertile was not significantly greater than that of the middle and highest BMD tertiles (1.4, SD 0.9) (*p* = 0.093) (Table 2).

### 3.3. Primary Outcomes

#### 3.3.1. Comparison of Age, BMI, and Nutrients’ Levels with Tertiles of Lumbar Spine BMD in Men and Women

As shown in Table 3, we observed that the proportion of low BMI in subjects with the highest BMD at the spine tended to be lower than that in subjects with the middle and lowest BMD for both genders (*p* = 0.009 in men, *p* < 0.001 in women). Men with the lowest lumbar spine BMD had a significantly greater prevalence (60.8%) of suboptimal plasma C compared to those with the middle (43.8%) and highest lumbar spine BMD (49.0%) (*p* < 0.001). However, there was no similar significant finding regarding vitamin C status with BMD in women (*p* = 0.520).

#### 3.3.2. The Correlations of the Lumbar Spine BMD Status with Age, BMI, and Nutrient Levels by Analysis of Logistic Regressions

The correlations of the spine BMD status with age, BMI, and the four nutrients’ levels for both genders are presented in Table 4.

We compared the lowest tertiles of lumbar spine BMD with the highest tertiles of BMD in variables of age, BMI, and nutrients’ levels using the method of logistic regressions. Univariate logistic regression in men identified three factors with statistical significance or marginal significance: suboptimal plasma vitamin C (≤8.8 mg/mL) (*p* = 0.004), suboptimal serum vitamin B12 (≤300 pg/mL) (*p* = 0.062), and low BMI (underweight/normal weight) (*p* = 0.011). In multivariate logistic regressions, the impacts of these factors remained to independently predict low BMD at the spine: suboptimal plasma vitamin C (adjusted OR 1.64, 95% CI 1.16, 2.31), suboptimal serum vitamin B12 (adjusted OR 2.05, 95% CI 1.02, 4.12), and low BMI (adjusted OR 1.68, 95% CI 1.12, 2.53).

In univariate logistic regression for women, only one factor (low BMI) was a significant predictor (crude OR: 3.36, 95% CI 2.11, 5.34), but suboptimal vitamin C and B12 were not. In multivariate logistic regressions, the factor of low BMI (underweight/normal weight) was the only independent predictor of the lowest lumbar spine BMD for women (AOR 3.42, 95% CI 2.13, 5.50). 

### 3.4. Secondary Outcomes

Distribution Differences of the Lumbar Spine BMD between Suboptimal and Sufficient Plasma Vitamin C in Men and Women.

As shown in Figure 1, plasma vitamin C levels in two categories (i.e., suboptimal (≤8.8 mg/L) and sufficient (>8.8 mg/L)) were positively associated with BMD at the spine in men (*p* = 0.004). However, no association was noted in women (*p* = 0.552) (Figure 1). 

## 4. Discussion

The current study was the first study to examine associations between the lumbar spine BMD and plasma vitamin C in young and early middle-aged adults without major chronic diseases, medications, and smoking [4,17,54], which may interfere with the interpretation of the associations between plasma vitamin C and BMD. We demonstrated that the prevalence of suboptimal plasma vitamin C in the subjects with the lowest tertile of the lumbar spine BMD was significantly higher than that in those with the middle and highest BMD in men. However, no association was noted in women. The specific findings were similar to a previous study reporting lower BMD at the hip associated with low serum ascorbic acid levels in men, but no association in premenopausal women [17]. Furthermore, it was found that plasma vitamin C levels in two categories (i.e., suboptimal (≤8.8 mg/L) and sufficient (>8.8 mg/L)) were positively associated with BMD at the spine in men (*p* < 0.001). However, no association was noted in women. Our results support the findings of Simon et al., who showed that there is a positive association of circulating vitamin C status in a concentration range from 1 to 9 mg/L with total proximal femoral BMD [17].

The main components of bone are organic matrix (90% type I collagen plus other non-collagenous proteins) and mineral matrix (hydroxyapatite crystal embedded in the collagen fibers). The degrees of mineralization play a major role in bone strength, while the organic matrix is primarily responsible for its toughness and adaptations during the individual’s lifetime [73,74,75]. In cell models, ascorbic acid can stimulate type I collagen synthesis [76]. In mice, vitamin C increases the Ca^2+^ deposition of a collagenous extracellular matrix in bone [9]. Vitamin C not only increases the expression of osteoblast differentiation genes and maintains the differentiated functions of osteoblasts [8,9], it also reduces the expression of osteoclast differentiation genes through Wnt family member 3A (Wnt3a)/β-catenin and mitogen-activated protein kinase signaling pathways [9]. Taken together, vitamin C can stimulate type I collagen formation, promote osteoblastogenesis, block osteoclastogenesis, and act as a mediator of bone matrix deposition, affecting both the quantity and quality of bone [8,9,10]. In this study, the relations between plasma vitamin C and BMD in men and women were conflicting. Plasma vitamin C had a significant impact on BMD in young and middle-aged men, whereas the relationship did not exist in young and middle-aged women. The reasons for the conflicting findings on the associations may be due to the following. First, compared to women, men have more androgen and less estrogen. Serum testosterone levels were found not to be associated with BMD in men [77]. Estrogen plays an important role for the prevention of osteoporosis in aged male rats [78]. Overall, serum estradiol is more related to BMD than testosterone in men [77]. In contrast, young and early middle-aged women with more endogenous estrogen [4] might not depend on the physiological protection of vitamin C as much as men. Second, the present study included Asians only. Race/ethnicity is an important factor affecting the associations between circulating vitamins and BMD [18]. Third, the prevalence of suboptimal circulating nutrients for any of the four selected nutrients in men was significantly greater than that in women. Men had a higher rate of nutrient deficiency in at least one nutrient. The synergistic effects between suboptimal plasma vitamin C and other nutrient deficiencies may have aggravating effects on bone health [20,25]. Further studies are needed to confirm the findings and clarify the complex associations between deficiencies of vitamin C/other nutrients and BMD. The present study is a cross-sectional study that cannot determine causality between vitamin C and its effects on bone in young and early middle-aged adults. Our findings suggest a need for an intervention study by taking foods enriched with vitamin C or supplements in young and middle-aged adults to assess the prevention effects on plasma vitamin C deficiency and osteoporosis. From the Framingham Osteoporosis Study (subjects’ mean age 75 years, SD 5), significant associations were found for dietary vitamin C (adjusting for supplemental vitamin C intake) in 4-year BMD changes at multiple bone sites among men, but not in women [79]. In this Framingham population, both men and women in the highest category of vitamin C intake, including diet and supplements, had significantly fewer hip fractures in a 17-year follow-up [80]. A study in California revealed that postmenopausal women who took vitamin C (100 to 5000 mg daily) plus calcium and estrogen had the highest BMD at multiple bone sites at an average 12.4-year follow-up [81]. Obviously, study design, sex, age, and other factors such as calcium and estrogen use have a great impact on the study results.

As expected, no osteoporotic fracture was found in our study population including 1455 adults aged 20–49 without major comorbidities. Osteoporosis often has no clinical sign. Bone pain can be an early sign of osteoporosis. Untreated or undertreated bone pain caused by osteoporosis can quickly lead to central sensitization, resulting in chronic osteoporotic pain [82]. However, it is not possible to assess the rate of bone pain in this study because of the lack of pain assessment in the database. Further large-scale studies are required to assess the rates of bone pain and osteoporotic fracture in young and middle-aged adults with or without major comorbidities.

Multivariate logistic regressions identified suboptimal serum vitamin B12 as an independent predictor for low BMD at the lumbar spine in men, whereas suboptimal serum folic acid was not found. In contrast, no correlation between BMD and serum folate or vitamin B12 levels was observed in women. Our findings were partly consistent with the results of the Framingham Osteoporosis Study [83], reporting that a lower BMD in men and women was significantly associated with low plasma B12. However, a study using data from the National Health and Nutrition Examination Survey in the United States (NHANES III) demonstrated no association between serum vitamin B12 and BMD, but there were positive associations of serum folate levels with BMD in both genders [18]. In a recent observational study in Taiwan, no correlations between bone loss and circulating folate/vitamin B12 concentrations were found in adults aged 40–65 [21]. Although molecular mechanisms regarding how low folate/vitamin B12 alone or in combination affects BMD have been demonstrated in cell models [19,20], the observational studies in the field of associations between vitamin B12/folate and BMD remain inconclusive. More studies are needed to clarify the associations, particularly in young and middle-aged adults.

In healthy young adults in the Netherlands, serum 25(OH)D levels are positively associated with BMD. Furthermore, serum 25(OH)D status is a determinant of peak BMD in both genders [23]. However, there was no association between unadjusted or adjusted serum 25(OH)D levels and BMD at the lumbar spine in this study. Our findings were in agreement with two studies showing no association between serum 25(OH)D concentrations and BMD in Japanese women aged 19–70 [84] and in American young physicians [85]. Among Chinese adults, serum 25(OH)D level is an important biomarker of BMD in elderly women, but not in young and middle-aged women [86]. In contrast, serum 25(OH)D levels are positively associated with lumbar spine BMD in middle-aged and elderly men, but not in young men [86]. Apparently, evidence from different populations and different ethnicities is inconsistent. The present study attempted to address the important public health problem—osteoporosis—which can be prevented if modifiable risk factors are identified and modified at an early stage. Both a long-term observation on a large population and a prolonged intervention in different subgroups are required to draw unequivocal conclusions.

Dietary patterns were not measured in our study and could not be adjusted in the models due to a lack of available food questionnaires’ information. Nonetheless, we discovered only 14.2% of men and 14.0% of women without suboptimal circulating nutrients for any of the four selected nutrients. There was a high prevalence of micronutrient deficiencies in adults who received health examinations at Chi Mei Medical Center, a 1200-bed tertiary referral center in Tainan with 22.5–22.9° N latitude on the southern Taiwan island (a subtropical region). Tainan city, with a population of over 1.8 million, is currently the fifth largest city in Taiwan, with the gross domestic product per capita of USD 25,000–28,000 in 2019–2020 [37]. The average plasma vitamin C level of the 1455 subjects in the present study was 9.3 mg/L (SD 3.2 mg/L), which was consistent with the mean (9.4 mg/L, SD 1.0 mg/L) of the healthy controls aged 50 years or older in a previous study in Taiwan [87]. A recent study in northern Taiwan revealed that the incidence rates of low plasma folate (<5.8 ng/mL) and low plasma B12 (<442 pg/mL) were 31.9% and 32.4%, respectively, in the adults aged 40–65 receiving health examinations [21]. The proportions of lower plasma folate and vitamin B12 in theirs and our results are similar. These results indicate that micronutrient deficiencies in Taiwan are common, not due to a selection bias in the current study. Importantly, these findings are consistent with a global report describing that an estimated one-third of people suffer from at least one form of micronutrient deficiency globally [88]. Apparently, micronutrient deficiency is still a public health problem.

Among the anthropometric variables, low BMI has been shown to be interrelated with BMD and the occurrence of osteoporosis in adults aged 50 and older [29,30]. In the current study, low BMI (underweight/normal weight, BMI < 23) was the only predictor for low BMD at the lumbar spine by multiple logistic regressions for both genders. Our findings provide support to previous studies reporting a positive correlation between BMI and BMD in both genders in young Brazilian cohorts [89] and middle-aged US adults [90]. However, other publications in this regard reported significant positive associations between BMI and BMD only in the elder groups (>60 years) for both genders, and no significant difference in BMD among young and middle-aged women and middle-aged men in US adults aged 20–90 years [91]. The literature is conflicting as to whether a positive association between BMI and BMD exists in young and middle-aged adults. Nevertheless, according to our findings in this study, body mass index seems to appear as a good indicator for the measurements of BMD in young and middle-aged adults.

There were some limitations in the current study. First, the study population was restricted to young and middle-aged Asians. Caution should be taken when generalizing the results to older or other populations because age and race/ethnicity are important factors for associations between vitamins and BMD [4,18]. Second, although the four selected circulating nutrients were examined, there may be additional nutrient deficiencies, such as vitamin A [18], which are associated with increased risks of low BMD. However, it is not possible to survey all nutrients related to BMD in one study. Third, this study was a retrospective cross-sectional study, which was not able to examine the causal relationship between vitamins and BMD. Causality between nutrients and their effects on bone can only be established in randomized controlled trials. Fourth, the present study did not include food frequency questionnaires or 24 h food recalls due to the original data lacking the information. However, circulating vitamins’ status provides more precise information of nutrients in the human body than that from food questionnaires [4,17]. Fifth, a single measure of circulating vitamins may not represent the long-term nutritional status. A well-designed cohort study is required to confirm the findings. Sixth, peak bone mass is contributed by physical activity, data of which can be obtained by a short questionnaire about physical activity at the time of evaluation [92]. The lack of physical activity data at the time of evaluation is a limitation of the present retrospective study. For avoiding possible bias, a well-designed cohort study is needed to further investigate the associations between physical activity, plasma vitamin C status, and longitudinal changes of BMD in the future.

## 5. Conclusions

The current study demonstrated that the prevalence of suboptimal plasma vitamin C among the subjects with the lowest tertile of the lumbar spine BMD was significantly higher than that in those with the middle and highest BMD in men, and that plasma vitamin C status in two categories (i.e., sufficient (>8.8 mg/L) and suboptimal (≤8.8 mg/L)) was positively associated with bone mineral density at the lumbar spine in men aged 20 to 49 years, but not in young and middle-aged women. Approximately 86% of the study population had suboptimal nutrients (nutritional insufficiency or deficiency) in at least one of the four selected nutrients. Nutrition is a modifiable risk factor for the prevention of bone loss [3,4,5,6]. The importance of a healthy diet and nutritional supplements should be emphasized in young and middle-aged adults. A well-designed cohort study is needed to examine the associations between plasma vitamin C status and longitudinal changes of bone mineral density.

## Figures and Tables

**Figure 1 nutrients-14-03556-f001:**
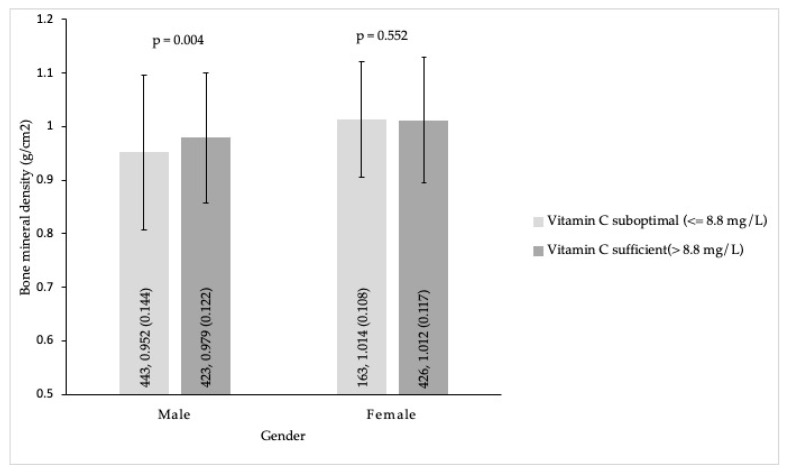
Bone mineral density (g/cm^2^) (number of cases, mean (standard deviation)) at the lumbar spine across two categories of plasma vitamin C (i.e., suboptimal and sufficient) in men and women. Mann–Whitney U-test was used for the test of the non-normally distributed continuous variables.

**Table 1 nutrients-14-03556-t001:** Demographic/anthropometric characteristics and laboratory and radiology results of the study population.

Variables	Male (*n* = 866)	Female (*n* = 589)	*p*
Age groups (years), median (IQR)	43 (6)	43 (6)	0.097
20–35, *n* (%)	92 (10.6%)	69 (11.7%)	0.515
36–49, *n* (%)	774 (89.4%)	520 (88.3%)	
Body height (kg), median (IQR)	172.1 (7.2)	159.6 (7.6)	<0.001
Body weight (cm), median (IQR)	74.5 (14.5)	55.1 (11.0)	<0.001
Body mass index (kg/m^2^), median (IQR)	25.1 (4.2)	21.5 (3.9)	<0.001
High BMI (≥23 kg/m^2^) (overweight/obesity), *n* (%)	671 (77.5%)	186 (31.6%)	<0.001
Low BMI (<23 kg/m^2^) (underweight/normal weight), *n* (%)	195 (22.5%)	403 (68.4%)	
Seasons			
Winter/Spring, *n* (%)	385 (44.5%)	301 (51.1%)	0.013
Summer/Autumn, *n* (%)	481 (55.5%)	288 (48.9%)	
Alcohol consumption	4 (0.4%)	2 (0.3%)	0.983
Osteoporotic fracture, *n* (%)	0	0	-
Bone mineral density at the lumbar spine (g/cm^2^), median (IQR)	0.953 (0.173)	1.011 (0.151)	<0.001
Plasma vitamin C (mg/L), median (IQR)	8.8 (4.2)	10.4 (3.8)	<0.001
Sufficient (>8.8 mg/L), *n* (%)	423 (48.8%)	426 (72.3%)	<0.001
Suboptimal (≤8.8 mg/L), *n* (%)	443 (51.2%)	163 (27.7%)	
Insufficient (6.1–8.8 mg/L), *n*	678	558	
Deficient (≤6.0 mg/L), *n*	188	31	
Serum 25(OH)D (ng/mL), median (IQR)	26.8 (10.6)	23.2 (9.2)	<0.001
Sufficient (≥30 ng/mL), *n* (%)	290 (33.5%)	108 (18.3%)	<0.001
Suboptimal (<30 ng/mL), *n* (%)	576 (66.5%)	481 (81.7%)	
Insufficient (20–30 ng/mL), *n*	710	405	
Deficient (<20 ng/mL), *n*	156	184	
Serum vitamin B12 (pg/mL), median (IQR)	509 (274)	621 (303)	<0.001
Sufficient (>300 pg/mL), *n* (%)	803 (92.7%)	569 (96.6%)	0.002
Suboptimal (≤300 pg/mL), *n* (%)	63 (7.3%)	20 (3.4%)	
Insufficient (200–300 pg/mL), *n*	858	587	
Deficient (<200 pg/mL), *n*	8	2	
Serum folic acid (ng/mL), median (IQR)	7.7 (5.1)	10.4 (5.1)	<0.001
Sufficient (>6 ng/mL), *n* (%)	619 (71.5%)	528 (89.6%)	<0.001
Suboptimal (≤6 ng/mL), *n* (%)	247 (28.5%)	61 (10.4%)	
Insufficient (3–6 ng/mL), *n*	860	588	
Deficient (<3 ng/mL), *n*	6	1	

25(OH)D: 25-hydroxyvitamin D; *n*: number of cases; IQR: interquartile range; SD: standard deviation; BMI: body mass index. -: Not calculated. Mann–Whitney U-test was used for comparison of the non-normally distributed continuous variables. Chi-square test was used for the differences in categorical variables among groups.

**Table 2 nutrients-14-03556-t002:** Distribution of suboptimal circulating nutrients for the four selected nutrients in the study population.

Subjects with Suboptimal Circulating Nutrients	None	1 Nutrient	2 Nutrients	3–4 Nutrients	*p*	Mean (SD)	*p*
Total (*n* = 1455)	192 (13.2%)	663 (45.6%)	427 (29.3%)	173 (11.9%)		1.4 (0.9)	
Gender, *n* (%)							
Men (*n* = 866), *n* (%)	117 (13.5%)	330 (38.1%)	275 (31.8%)	144 (16.6%)	<0.001	1.5 (1.0)	<0.001
Women (*n* = 589), *n* (%)	75 (12.7%)	333 (56.5%)	152 (25.8%)	29 (4.9%)		1.2 (0.7)	
Age groups							
20–35 (*n* = 161), *n* (%)	10 (6.2%)	67 (41.6%)	46 (28.6%)	38 (23.6%)	<0.001	1.7 (0.9)	<0.001
36–49 (*n* = 1294), *n* (%)	182 (14.1%)	596 (46.1%)	381 (29.4%)	135 (10.4%)		1.4 (0.9)	
Lumbar spine BMD							
The lowest BMD (*n* = 483), *n* (%)	53 (11.0%)	223 (46.2%)	144 (29.8%)	63 (13.0%)	0.290	1.5 (0.9)	0.093
The middle and highest (*n* = 972), *n* (%)	141 (14.5%)	436 (44.9%)	282 (29.0%)	113 (11.6%)		1.4 (0.9)	

*n*: Number of cases; SD: standard deviation. Chi-square test for the differences in categorical variables among groups. Mann–Whitney U-test was used for comparison of the non-normally distributed continuous variables.

**Table 3 nutrients-14-03556-t003:** (**a**) Comparison of age, BMI, and nutrients’ levels with tertiles of lumbar spine BMD in men. (**b**) Comparison of age, BMI, and nutrients’ levels with tertiles of lumbar spine BMD in women.

	**(a)**			
	**Bone Mineral Density of Lumbar Spine in Men**			
**Variables (*n* = 866)**	**Lowest (*n* = 286)**	**Middle (*n* = 288)**	**Highest (*n* = 292)**	* **p** *
Age groups (years)				0.271
20–35, *n* (%)	37 (12.9%)	29 (10.1%)	26 (8.9%)	
36–49, *n* (%)	249 (87.1%)	259 (89.9%)	266 (91.1%)	
Body mass index (kg/m^2^)				0.009
High BMI (≥23 kg/m^2^) (overweight/obesity), *n* (%)	214 (74.8%)	213 (74.0%)	244 (83.6%)	
Low BMI (<23 kg/m^2^) (underweight/normal weight), *n* (%)	72 (25.2%)	75 (26.0%)	48 (16.4%)	
Vitamin C (mg/L)				<0.001
Sufficient (>8.8 mg/L), *n* (%)	112 (39.2%)	162 (56.3%)	149 (51.0%)	
Suboptimal (≤8.8 mg/L), *n* (%)	174 (60.8%)	126 (43.8%)	143 (49.0%)	
Serum 25(OH)D (ng/mL)				0.806
Sufficient (≥30 ng/mL), *n* (%)	100 (35.0%)	95 (33.0%)	95 (32.5%)	
Suboptimal (<30 ng/mL), *n* (%)	186 (65.0%)	193 (67.0%)	197 (67.5%)	
Vitamin B12 (pg/mL)				0.114
Sufficient (>300 pg/mL), *n* (%)	261 (91.3%)	263 (91.3%)	278 (95.2%)	
Suboptimal (≤300 pg/dL), *n* (%)	25 (8.7%)	25 (8.7%)	14 (4.8%)	
Folic acid (ng/mL)				0.426
Sufficient (>6 ng/mL), *n* (%)	201 (70.3%)	214 (74.3%)	204 (69.9%)	
Suboptimal (≤6.0 ng/mL), *n* (%)	85 (29.7%)	74 (25.7%)	88 (30.1%)	
Alcohol consumption				-
−	284 (99.3%)	288 (100.0%)	290 (99.3%)	
+	2 (0.7%)	0 (0%)	2 (0.7%)	
	**(b)**			
	**Bone Mineral Density of Lumbar Spine in Women**			
**Variables (*n* = 589)**	**Lowest (*n* = 197)**	**Middle (*n* = 197)**	**Highest (*n* = 195)**	* **p** *
Age groups (years)				0.724
20–35, *n* (%)	22 (11.2%)	26 (13.2%)	21 (10.8%)	
36–49, *n* (%)	175 (88.8%)	171 (86.8%)	174 (89.2%)	
Body mass index (kg/m^2^)				<0.001
High BMI (≥23 kg/m^2^) (overweight/obesity), *n* (%)	35 (17.8%)	65 (33.0%)	85 (43.6%)	
Low BMI (<23 kg/m^2^) (underweight/normal weight), *n* (%)	161 (81.7%)	132 (67.0%)	110 (56.4%)	
Vitamin C (mg/L)				0.520
Sufficient (>8.8 mg/L), *n* (%)	148 (75.1%)	138 (70.1%)	140 (71.8%)	
Suboptimal (≤8.8 mg/L), *n* (%)	49 (24.9%)	59 (29.9%)	55 (28.2%)	
Serum 25(OH)D (ng/mL)				0.763
Sufficient (≥30 ng/mL), *n* (%)	39 (19.8%)	36 (18.3%)	33 (16.9%)	
Suboptimal (<30 ng/mL), *n* (%)	158 (80.2%)	161(81.7%)	162 (83.1%)	
Vitamin B12 (pg/mL)				0.317
Sufficient (>300 pg/mL), *n* (%)	190 (96.4%)	187 (94.9%)	191 (97.9%)	
Suboptimal (≤300 pg/dL), *n* (%)	7 (3.6%)	10 (5.1%)	4 (2.1%)	
Folic acid (ng/mL)				0.271
Sufficient (>6 ng/mL), *n* (%)	175 (88.8%)	182 (92.4%)	171 (87.7%)	
Suboptimal (≤6.0 ng/mL), *n* (%)	22 (11.2%)	15 (7.6%)	24 (12.3%)	
Alcohol consumption				-
−	196 (99.5%)	196 (99.5%)	195 (100%)	
+	1 (0.5%)	1 (0.5%)	0 (0%)	

25(OH)D: 25-hydroxyvitamin D; *n*: number of cases; SD: standard deviation; BMI: body mass index. -: Not calculated. ANOVA was used for the differences in continuous data among groups. Chi-square test was used for the differences in categorical variables among groups. −: less than once a month consumption; +: once or more a month consumption.

**Table 4 nutrients-14-03556-t004:** (**a**) The correlations of the lumbar spine BMD status with age, BMI, and nutrients’ levels by the method of logistic regressions in men. (**b**) The correlations of the lumbar spine BMD status with age, BMI, and nutrients’ levels by the method of logistic regressions in women.

		**(a)**				
**Men**	**Lowest BMD (*n* = 286)**	**Highest BMD (*n* = 292)**	**Crude OR (95% CI)**	* **p** *	**Adjusted OR (95% CI)**	* **p** *
Age groups (years)						
20–35, *n* (%)	37 (58.73)	26 (41.27)	1.0		1.0	
36–49, *n* (%)	249 (48.35)	266 (51.65)	0.66 (0.39–1.12)	0.122	0.65 (0.38–1.14)	0.132
Body mass index (kg/m^2^)						
High BMI (≥23 kg/m^2^) (overweight/obesity), *n* (%)	211 (46.68)	241 (53.32)	1.0		1.0	
Low BMI (<23 kg/m^2^) (underweight/normal weight), *n* (%)	75 (59.52)	51 (40.48)	1.68 (1.13–2.51)	0.011	1.68 (1.12–2.53)	0.012
Vitamin C (mg/L)						
Sufficient (>8.8 mg/L), *n* (%)	112 (42.91)	149 (57.09)	1.0		1.0	
Suboptimal (≤8.8 mg/L), *n* (%)	174 (54.89)	143 (45.11)	1.62 (1.16–2.25)	0.004	1.64 (1.16–2.31)	0.004
Serum 25 (OH)D (ng/mL)						
Sufficient (≥30 ng/mL), *n* (%)	100 (51.28)	95 (48.72)	1.0		1.0	
Suboptimal (<30 ng/mL), *n* (%)	186 (48.56)	197 (51.44)	0.90 (0.64–1.27)	0.537	0.91 (0.64–1.30)	0.611
Vitamin B12 (pg/mL)						
Sufficient (>300 pg/mL), *n* (%)	261 (48.42)	278 (51.58)	1.0		1.0	
Suboptimal (≤300 pg/dL), *n* (%)	25 (64.10)	14 (35.90)	1.90 (0.97–3.74)	0.062	2.05 (1.02–4.12)	0.043
Folic acid (ng/mL)						
Sufficient (>6 ng/mL), *n* (%)	201 (49.63)	204 (50.37)	1.0		1.0	
Suboptimal (≤6.0 ng/mL), *n* (%)	85 (49.13)	88 (50.87)	0.98 (0.69–1.40)	0.912	0.78 (0.53–1.14)	0.204
		**(b)**				
**Women**	**Lowest BMD (*n* = 197)**	**Highest BMD (*n* = 195)**	**Crude OR (95% CI)**	** *p* **	**Adjusted OR (95% CI)**	** *p* **
Age groups (years)						
20–35, *n* (%)	22 (51.16)	21 (48.84)	1.0		1.0	
36–49, *n* (%)	175 (50.14)	174 (49.86)	0.96 (0.51–1.81)	0.899	1.17 (0.61–2.27)	0.634
Body mass index (kg/m^2^)						
High BMI (≥23 kg/m^2^) (overweight/obesity), *n* (%)	35 (29.91)	82 (70.09)	1.0		1.0	
Low BMI (<23 kg/m^2^) (underweight/normal weight), *n* (%)	162 (58.91)	113 (41.09)	3.36 (2.11–5.34)	<0.001	3.42 (2.13–5.50)	<0.001
Vitamin C (mg/L)						
Sufficient (>8.8 mg/L), *n* (%)	148 (51.39)	140 (48.61)	1.0		1.0	
Suboptimal (≤8.8 mg/L), *n* (%)	49 (47.12)	55 (52.88)	0.82 (0.54–1.32)	0.456	0.87 (0.54–1.39)	0.552
Serum 25 (OH)D (ng/mL)						
Sufficient (≥30 ng/mL), *n* (%)		33 (45.83)	1.0		1.0	
Suboptimal (<30 ng/mL), *n* (%)	158 (49.38)	162 (50.63)	0.83 (0.49–1.38)	0.464	1.01 (0.59–1.73)	0.959
Vitamin B12 (pg/mL)						
Sufficient (>300 pg/mL), *n* (%)	190 (49.87)	191(50.13)	1.0		1.0	
Suboptimal (≤300 pg/dL), *n* (%)	7 (63.64)	4 (36.36)	1.76 (0.51–6.11)	0.374	2.01 (0.55–7.39)	0.295
Folic acid (ng/mL)						
Sufficient (>6 ng/mL), *n* (%)	175 (50.58)	171 (49.42)	1.0		1.0	
Suboptimal (≤6.0 ng/mL), *n* (%)	22 (47.83)	24 (52.17)	0.90 (0.48–1.66)	0.726	0.99 (0.51–1.92)	0.966

25(OH)D: 25-hydroxyvitamin D; *n*: number of cases; SD: standard deviation; OR: odds ratio; BMI: body mass index. Adjusted ORs were adjusted variables, including age, BMI, vitamin C, 25(OH)D, vitamin B12, folic acid, and season.

## Data Availability

Anonymized data not published within this article will be made available and shared upon request from any qualified investigator.

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
