# Peer review of "Suboptimal Plasma Vitamin C Is Associated with Lower Bone Mineral Density in Young and Early Middle-Aged Men: A Retrospective Cross-Sectional Study"

_nutrients, 2022, doi:10.3390/nu14173556_

Round 1
Reviewer 1 Report
The Authors of this article aim to evaluate associations between bone mineral density (BMD) and four selected circulating nutrients, particularly vitamin C, among young and middle-aged adults (20-49 years). As the authors said, finding reversible factors to prevent osteoporosis is a topic of great importance, and the current literature on this topic is still discordant.
However, some issues should be considered:
Paragraph Introduction:
- Page 2, line 46 the verbs “increased and persisted” should be replaced with “increase.”
- Page 2, line 51 the verb “was” should be replaced with “i.”
- Page 2, lines 51-52 the Authors report that up to 40% of peak bone mass is contributed by environmental factors including lifestyles, physical activity, and nutrition. It would have been interesting if they added physical activity as a variable in the study. This way they could have compared it with the others and avoided a possible bias. Since it is a retrospective study, this could have been accomplished by calling the patients and submitting a short questionnaire about physical activity at the time of evaluation.
- Page 2, line 92 the word “associations” should be singular not plural.
- Page 3, lines 106-107 The Authors state that low BMD was defined as BMD in the lowest tertile of the lumbar spine BMD. It is not clear why the Authors used the lowest tertile as a pathological reference instead of the T score and Z score, which are usually used to identify objective osteoporosis.
- Page 3, lines 111-113 As describing the primary outcome the Authors state that they want to identify risk factors for low BMD. It would be more appropriate to say that they want to explore possible associations since this is a transversal stud. As such, it is impossible to determine causality between nutrients and their effects on bone.
Paragraph Material and Methods:
- Page 3, line 131 I would re-formulate the sentence in “participants who had the DXA examination”.
-Page 3, Exclusion Criteria The Authors excluded patients with major chronic diseases and smokers from the study to avoid interfering with the interpretation of the associations between the nutrients and BDM. Why weren’t medications (e.i. corticosteroids, some antiepileptic and psychiatric drugs) and other diseases (such as hyperthyroidism, hyperparathyroidism, Cushing sdr, hypogonadism, rheumatological diseases) taken into consideration as exclusion criteria? They could alter BMD and if not detected could be a bias for the study.
- Page 5, Statistical Analysis The Authors state that the difference of continuous data between men and women was conducted by Student t-test. The Authors should specify whether the data had a normal distribution, if not, they should normalize data using either Shapiro-Wilk or Kolmogorov-Smirnov tests.
- Page 5, lines 229-232 Secondary outcomes were to explore associations between BMD and three categories of plasma vitamin C as well as BMI. To do so, they conducted an analysis of variance (ANOVA): it is not clear though if the Barlett test was executed in order to establish homoscedasticity. I suggest the Authors clarify this point.
Tables:
- In table 4b there is an error, the values of high and low BMI are reversed, it should be: high BMI ≧ 23 kg/m2, low BMI <23 kg/m2;
Paragraph Results:
Among the anthropometric variables, low BMI has been shown to be interrelated with BMD and the occurrence of osteoporosis in adults. In this study, the authors also found this correlation, as the only predictor for low BMD by multiple logistic regressions for both genders. Having that said, I suggest that the Authors should use BMI as a variable in the adjusted odds ratio (AOR). In the Results, the authors state that in multivariate logistic regressions, the impacts of suboptimal vitamin C, suboptimal vitamin B12, and low BMI remained to independently predict low BMD at the spine: the AOR though is only adjusted for age groups, season, and alcohol consumption. It would be interesting to see if the impacts of the studied variables would remain the same with an AOR adjusted for BMI.
Paragraph Discussion:
In the discussion, the authors conclude that, considering that the analysis of the data showed that men had higher rates of nutrient deficiency which may aggravate the impact of suboptimal plasma vitamin C on bone health, taking foods enriched with vitamin C or supplements seemed crucial for preventing osteoporosis in young and middle-aged men (page 12 lines 364-367). This statement cannot be made based on this study since it is a transversal study and cannot determine causality between nutrients and their effects on bone.
Author Response
We would like to express our gratitude to the Reviewer for the time taken to review our manuscript and the professional comments that substantially improved the quality of our work. Please find our point-by-point response below. Please also kindly note that the corresponding changes made in the revised manuscript are marked in red.
The Authors of this article aim to evaluate associations between bone mineral density (BMD) and four selected circulating nutrients, particularly vitamin C, among young and middle-aged adults (20-49 years). As the authors said, finding reversible factors to prevent osteoporosis is a topic of great importance, and the current literature on this topic is still discordant.
However, some issues should be considered:
Paragraph Introduction:
- Page 2, line 46 the verbs “increased and persisted” should be replaced with “increase.”
Response:
Thanks for the correction. The phrase is corrected as suggested.
- Page 2, line 51 the verb “was” should be replaced with “is.”
Response:
Thanks for the correction. The verb used is corrected as suggested.
- Page 2, lines 51-52 the Authors report that up to 40% of peak bone mass is contributed by environmental factors including lifestyles, physical activity, and nutrition. It would have been interesting if they added physical activity as a variable in the study. This way they could have compared it with the others and avoided a possible bias. Since it is a retrospective study, this could have been accomplished by calling the patients and submitting a short questionnaire about physical activity at the time of evaluation.
Response:
We thank the reviewer for the insightful comment. The following has been added in “Limitation”.
Sixth, peak bone mass is contributed by physical activity, data of which can be obtained by a short questionnaire about physical activity at the time of evaluation. (The BPAQ: a bone-specific physical activity assessment instrument, 2008) Lack of the physical activity data at the time of evaluation is a limitation of the present retrospective study. For avoiding possible bias resulting, a well-designed cohort study is needed to further investigate the associations between physical activity, plasma vitamin C status and longitudinal changes of BMD in the future.
- Page 2, line 92 the word “associations” should be singular not plural.
Response:
Thanks for the correction. The misspelled word is corrected as suggested.
- Page 3, lines 106-107 The Authors state that low BMD was defined as BMD in the lowest tertile of the lumbar spine BMD. It is not clear why the Authors used the lowest tertile as a pathological reference instead of the T score and Z score, which are usually used to identify objective osteoporosis.
Response:
Regarding the major limitations that the Reviewer insightfully pointed out, we have made corresponding amendments in the revised manuscript
Bone mineral density measured by DXA is a direct method and a good indicator of the bone health. A T-score is a standard deviation showing the difference between the BMD of a subject and the average of the healthy 30 year old adults. It is not used commonly for the study including young ages. (Change in bone mineral density as a function of age in women and men and association with the use of antiresorptive agents, 2008) A Z-score compares the bone density to the average values for a person of the same age and gender. A low Z-score (below -2.0) is a warning sign for less bone mass than expected for the age. There were only 8.8% of men and 2.0% of women to have Z-score below -2.0 in our study population. Therefore, BMD was tertiled to identify a susceptible population. (Association of Renal Function and Menopausal Status with Bone Mineral Density in Middle-aged Women, 2015; A cross-sectional association between bone mineral density and parathyroid hormone and other biomarkers in community-dwelling young adults: the CARDIA study, Journal of Clinical Endocrinology & Metabolism, 2013)
- Page 3, lines 111-113 As describing the primary outcome the Authors state that they want to identify risk factors for low BMD. It would be more appropriate to say that they want to explore possible associations since this is a transversal stud. As such, it is impossible to determine causality between nutrients and their effects on bone.
Response:
Thanks for the reviewer’s insightful comment. The term “risk factors” is replaced by “possible associations”.
Paragraph Material and Methods:
- Page 3, line 131 I would re-formulate the sentence in “participants who had the DXA examination”.
Response:
Thanks for the correction. The sentence is corrected as suggested.
-Page 3, Exclusion Criteria The Authors excluded patients with major chronic diseases and smokers from the study to avoid interfering with the interpretation of the associations between the nutrients and BDM. Why weren’t medications (e.i. corticosteroids, some antiepileptic and psychiatric drugs) and other diseases (such as hyperthyroidism, hyperparathyroidism, Cushing sdr, hypogonadism, rheumatological diseases) taken into consideration as exclusion criteria? They could alter BMD and if not detected could be a bias for the study.
Response:
We are grateful to the Reviewer for the professional comments. We have added these comorbidities and medications in methodology section.
….(3) subjects who had diagnostic codes of human immunodeficiency virus infection (ICD-10 B20-B24)[1], solid organ transplantation (ICD-10 Z94)[2], chronic liver diseases (ICD-10 K70-77)[3] and/or chronic kidney disease (ICD-10 N18.5, N18.6 or the estimated glomerular filtration rate (eGFR) <60 ml/min/1.73 m2)[4], hyperthyroidism (ICD-10 E05)[5], hyperparathyroidism (ICD-10 E21)[6], Cushing’s syndrome (ICD-10 E24)[7], hypogonadism (ICD-10 E29.1)[8], rheumatological diseases (ICD-10 M05, M32-35)[9] which are potential confounding factors in the relationship of BMD and vitamin status; (4) participants who took medications (i.e. corticosteroids [Anatomical Therapeutic Chemical classification system (ATC)-D07][10], some antiepileptic (ATC-N03)[11] and psychiatric drugs (ATC-05A)[12]) were excluded because these medications affect bone metabolism;….
- Page 5, Statistical Analysis The Authors state that the difference of continuous data between men and women was conducted by Student t-test. The Authors should specify whether the data had a normal distribution, if not, they should normalize data using either Shapiro-Wilk or Kolmogorov-Smirnov tests.
Response:
According to the reviewer’s comment, we do the normality using Kolmogorov-Smirnov test. Due to all variables did not present the normal distribution, the Mann-Whitney U test was used to estimate the difference between males and females. The information has been incorporated into the Methods, and Table 1 of the revised manuscript.
The following is the results of the supplemental test for the normality using Kolmogorov-Smirnov test. This table was not included in the revised manuscript due to space limitations.
Supplemental test for the normality using Kolmogorov-Smirnov test
|
Male |
Female |
Age (years) |
<0.0100 |
<0.0100 |
Body height (kg) |
0.0779 |
0.0125 |
Body weight (cm) |
<0.0100 |
<0.0100 |
Body mass index (kg/m2) |
<0.0100 |
<0.0100 |
Bone mineral density at the lumbar spine (g/cm2) |
<0.0100 |
0.1500 |
Plasma vitamin C (mg/L) |
<0.0100 |
0.1500 |
Serum 25(OH)D (ng/mL) |
<0.0100 |
<0.0100 |
Serum vitamin B12 (pg/mL) |
<0.0100 |
<0.0100 |
Serum folic acid (ng/mL) |
<0.0100 |
<0.0100 |
- Page 5, lines 229-232 Secondary outcomes were to explore associations between BMD and three categories of plasma vitamin C as well as BMI. To do so, they conducted an analysis of variance (ANOVA): it is not clear though if the Barlett test was executed in order to establish homoscedasticity. I suggest the Authors clarify this point.
Response:
Thanks for the reviewer's important comment. We made a clarification that ANOVA in here was used to present the distribution difference of the BMD at spine between three categories of plasma vitamin C (i.e., deficient, insufficient and sufficient) in men and women, respectively. However, we modified the design from three categories to two categories of plasma vitamin C to make the findings easier to understand.
The updated description in Method of the revised manuscript is as below,
"In addition, the distribution difference between the BMD at spine and two categories of plasma vitamin C [i.e., suboptimal (<8.8 mg/L) and sufficient (>8.8 mg/L)] in males and females were conducted using the Mann–Whitney U-test if non-normality distribution after Kolmogorov-Smirnov testing.”
Tables:
- In table 4b there is an error, the values of high and low BMI are reversed, it should be: high BMI ≧ 23 kg/m2, low BMI <23 kg/m2;
Response:
Thanks for the correction. The table description is corrected as suggested.
Paragraph Results:
Among the anthropometric variables, low BMI has been shown to be interrelated with BMD and the occurrence of osteoporosis in adults. In this study, the authors also found this correlation, as the only predictor for low BMD by multiple logistic regressions for both genders. Having that said, I suggest that the Authors should use BMI as a variable in the adjusted odds ratio (AOR). In the Results, the authors state that in multivariate logistic regressions, the impacts of suboptimal vitamin C, suboptimal vitamin B12, and low BMI remained to independently predict low BMD at the spine: the AOR though is only adjusted for age groups, season, and alcohol consumption. It would be interesting to see if the impacts of the studied variables would remain the same with an AOR adjusted for BMI.
Response:
Thanks for the important comments.
In Table 4, adjusted OR were adjusted the listed variables, including age, BMI, vitamin C, 25(OH)D, vitamin B12, folic acid, and season.
Paragraph Discussion:
In the discussion, the authors conclude that, considering that the analysis of the data showed that men had higher rates of nutrient deficiency which may aggravate the impact of suboptimal plasma vitamin C on bone health, taking foods enriched with vitamin C or supplements seemed crucial for preventing osteoporosis in young and middle-aged men (page 12 lines 364-367). This statement cannot be made based on this study since it is a transversal study and cannot determine causality between nutrients and their effects on bone.
Response:
In response to the Reviewer’s insightful comment, we have reviewed the literature for the information pointed out by the Reviewer. The statement has been changed.
Men had a higher rate of nutrient-deficiency in at least one nutrient. The synergistic effects between suboptimal plasma vitamin C and other nutrient deficiency may have aggravating effects on bone health (20, 25). Further studies are required to confirm these findings and clarify the complex associations between deficiency of vitamin C/other nutrients and BMD. ….
……………………………..The present study is a cross-sectional study that cannot determine causality between vitamin C and their effects on bone in adult young and early middle aged men and women. Our findings suggest a need for an intervention study by taking foods enriched with vitamin C or supplements in young and middle-aged adults to prevent plasma vitamin C deficiency and assess the prevention effects on osteoporosis.
- Goh, S.S.L., et al., Reduced bone mineral density in human immunodeficiency virus-infected individuals: a meta-analysis of its prevalence and risk factors: supplementary presentation. Osteoporos Int, 2018. 29(7): p. 1683.
- Lan, G.B., et al., Current Status of Research on Osteoporosis after Solid Organ Transplantation: Pathogenesis and Management. Biomed Res Int, 2015. 2015: p. 413169.
- Jeong, H.M. and D.J. Kim, Bone Diseases in Patients with Chronic Liver Disease. Int J Mol Sci, 2019. 20(17).
- Bover, J., et al., Osteoporosis, bone mineral density and CKD-MBD: treatment considerations. J Nephrol, 2017. 30(5): p. 677-687.
- Delitala, A.P., A. Scuteri, and C. Doria, Thyroid Hormone Diseases and Osteoporosis. J Clin Med, 2020. 9(4).
- Lewiecki, E.M. and P.D. Miller, Skeletal effects of primary hyperparathyroidism: bone mineral density and fracture risk. J Clin Densitom, 2013. 16(1): p. 28-32.
- Mazziotti, G., S. Frara, and A. Giustina, Pituitary Diseases and Bone. Endocr Rev, 2018. 39(4): p. 440-488.
- Rochira, V., Late-onset Hypogonadism: Bone health. Andrology, 2020. 8(6): p. 1539-1550.
- Pereira, R.M., J.F. Carvalho, and E. Canalis, Glucocorticoid-induced osteoporosis in rheumatic diseases. Clinics (Sao Paulo), 2010. 65(11): p. 1197-205.
- Buckley, L. and M.B. Humphrey, Glucocorticoid-Induced Osteoporosis. N Engl J Med, 2018. 379(26): p. 2547-2556.
- Beniczky, S.A., et al., Bone mineral density in adult patients treated with various antiepileptic drugs. Seizure, 2012. 21(6): p. 471-2.
- Crews, M.P. and O.D. Howes, Is antipsychotic treatment linked to low bone mineral density and osteoporosis? A review of the evidence and the clinical implications. Hum Psychopharmacol, 2012. 27(1): p. 15-23.

Reviewer 2 Report
The aim of this retrospective observational study is to “evaluate associations between bone mineral density (BMD) and four selected circulating nutrients, particularly vitamin C among adults aged 20-49 years.” The study follows a retrospective cross-sectional design and uses different statistic models including regression analyses. The authors conclude that vitamin C levels were positively associated with the lumbar spine BMD in young and early middle-aged men and that a well-designed cohort study is needed to confirm the findings.
The limitations of the study design are discussed. However, it should be pointed out more clearly that the study is generating several hypotheses.
To inform future cohort studies can the authors please consider the following comments:
- The authors state that (p1, line 42): “osteoporosis has no clinical manifestations until there is a fracture”. It is recognised that bone pain can be an early sign of osteoporosis. Can the authors comment on this and could information on fracture rate and bone pain be included in the analysis?
- The authors state (p3, line 126) that the analysis of vitamins B12, C and D and of folic acid is not insured. This could be a selection bias and impacts on the interpretation and comparability of the study results. More specific information on the study population, region, climate and local factors needs to be included.
- The high prevalence of nutrient deficiencies might highlight a poor diet. Can the authors please comment on the overall nutritional status in their population and discuss potential reasons for this finding.
- Activity is an important factor impacting on an individual’s risk of osteoporosis. Data are not included. This is a major limitation. Can you please discuss.
- The authors present a thorough review of the literature on the impact of vitamin C on osteoporosis. On page 2, line 68 they state “Evidence in the field of associations between vitamin C intake and BMD is inconsistent across studies”. Can they please discuss this in view of their findings? Are there studies on sex prevalences of vitamin deficiencies?
- The rationale for the categorisation of the BMD findings is unclear. For characterisation expressing BMD values in relation to a standard population (eg in T-scores) would be important. Can these be added?
- The authors state (p12, line 357) that “compared to women, men have less estrogen which plays an important role for the prevention of osteoporosis”. Can this section be expanded and can they also comment on the impact of testosterone?
- The authors also state (p12, line 366) that “, taking foods enriched with vitamin C or supplements seems 366 crucial for preventing osteoporosis in young and middle-age men”. Please expand the discussion on this topic and whether they are any trial in this area.
- There are missing p-values in table 3. Can the authors please explain this and add the values if required?
There are minor spelling errors:
- P3, line 140: “has” should be “have”
- P9, line 291; “nutritents” should be nutrients
Author Response
We would like to express our gratitude to the Reviewer for the time taken to review our manuscript and the professional comments that substantially improved the quality of our work. Please find our point-by-point response below. Please also kindly note that the corresponding changes made in the revised manuscript are marked in green.
The aim of this retrospective observational study is to “evaluate associations between bone mineral density (BMD) and four selected circulating nutrients, particularly vitamin C among adults aged 20-49 years.” The study follows a retrospective cross-sectional design and uses different statistic models including regression analyses. The authors conclude that vitamin C levels were positively associated with the lumbar spine BMD in young and early middle-aged men and that a well-designed cohort study is needed to confirm the findings.
The limitations of the study design are discussed. However, it should be pointed out more clearly that the study is generating several hypotheses.
To inform future cohort studies can the authors please consider the following comments:
- The authors state that (p1, line 42): “osteoporosis has no clinical manifestations until there is a fracture”. It is recognised that bone pain can be an early sign of osteoporosis. Can the authors comment on this and could information on fracture rate and bone pain be included in the analysis?
Response:
We appreciate the reviewer’s help on improving the readability of our paper. The following has been added in “Discussion” of the revised manuscript.
As expected, no osteoporotic fracture was found in our study population including 1455 adults aged 20-49 without major comorbidities. Osteoporosis often has no clinical sign. Bone pain can be an early sign of osteoporosis. Untreated or undertreated bone pain caused by osteoporosis can quickly lead to central sensitization resulting in chronic osteoporotic pain[82]. However, it is not possible to assess the rate of bone pain in this study because of lacking of pain assessment in the database. Further large-scale studies are required to assess the rates of bone pain and osteoporotic fracture in young and middle aged adults with or without major comorbidities.
- The authors state (p3, line 126) that the analysis of vitamins B12, C and D and of folic acid is not insured. This could be a selection bias and impacts on the interpretation and comparability of the study results. More specific information on the study population, region, climate and local factors needs to be included.
- The high prevalence of nutrient deficiencies might highlight a poor diet. Can the authors please comment on the overall nutritional status in their population and discuss potential reasons for this finding.
Response:
We thank the reviewer for the insightful comment. The following has been added in “Discussion” of the revised manuscript.
Dietary patterns were not measured in our study and could not be adjusted in the models due to lacking of available food questionnaires information. Nonetheless, we discovered only 14.2% of men and 14.0% of women without suboptimal circulating nutrients for any of the four selected nutrients. There was a high prevalence of micronutrient deficiencies in adults who received health examinations at Chi Mei Medical Center, a 1200-bed tertiary referral center in Tainan with 22.5–22.9〫N latitude on the southern Taiwan island (a subtropical region). Tainan city with a population of over 1.8 million is currently the fifth largest city in Taiwan with the gross domestic product per capita USD 25,000–28,000 in 2019–2020[37]. A recent study in northern Taiwan revealed that the incidence rates of low plasma folate (<5.8 ng/mL) and low plasma B12 (<442 pg/mL) were 31.9% and 32.4% respectively in the adults aged 40-65 receiving health examinations[21] similar. These results indicate that micronutrients deficiencies in Taiwan are common not due to a selection bias in the current study. Importantly, these findings are consistent with a global report describing an estimated one-third of people suffering from at least one form of micronutrient deficiency globally[87]. Apparently, micronutrient deficiency is still a public health problem.
- Activity is an important factor impacting on an individual’s risk of osteoporosis. Data are not included. This is a major limitation. Can you please discuss.
Response:
Regarding the major limitations that the Reviewer insightfully pointed out, we have made corresponding amendments in “Limitation” of the revised manuscript.
Sixth, peak bone mass is contributed by physical activity, data of which can be obtained by a short questionnaire about physical activity at the time of evaluation[91]. Lack of the physical activity data at the time of evaluation is a limitation of the present retrospective study. For avoiding possible bias resulting, a well-designed cohort study is needed to further investigate the associations between physical activity, plasma vitamin C status and longitudinal changes of BMD in the future.
- The authors present a thorough review of the literature on the impact of vitamin C on osteoporosis. On page 2, line 68 they state “Evidence in the field of associations between vitamin C intake and BMD is inconsistent across studies”. Can they please discuss this in view of their findings? Are there studies on sex prevalences of vitamin deficiencies?
Response:
The Reviewer’s professional concerns are sincerely appreciated. Accordingly, we have included the following in “Discussion”.
From the Framingham Osteoporosis Study with subjects (mean age 75 years, SD 5), significant associations were found for dietary vitamin C (adjusting for supplemental vitamin C intake) in 4-y BMD changes at multiple bone sites among men, but not in women[79]. In this Framingham population, both of men and women in the highest category of vitamin C intake including diet and supplements had significantly fewer hip fractures in a 17-year follow-up[80]. A study in California revealed that postmenopausal women who took vitamin C (100 to 5,000 mg daily) plus calcium and estrogen had the highest BMD at multiple bone sites in an average 12.4-year follow-up[81]. Obviously, study design, sex, age and other factors such as calcium and estrogen use display a great impact on the study results.
- The rationale for the categorisation of the BMD findings is unclear. For characterisation expressing BMD values in relation to a standard population (eg in T-scores) would be important. Can these be added?
Response:
We agree with the reviewer’ comment that we need to clarify the rationale for the categorisation of the BMD. As required, the following sentences were inserted to make clear.
Bone mineral density measured by DXA is a direct method and a good indicator of the bone health. A T-score is a standard deviation showing the difference between the BMD of a subject and the average of the healthy 30 year old adults. Thus, BMD not T-score is often used in the study including the youth. (Change in bone mineral density as a function of age in women and men and association with the use of antiresorptive agents, 2008) A Z-score compares the bone density to the average values for a person of the same age and gender. A low Z-score (below -2.0) is a warning sign for less bone mass than expected for the age. There were only 8.8% of men and 2.0% of women to have Z-score below -2.0 in our study population. Therefore, BMD was tertiled to identify a susceptible population in men and women, respectively. (Association of Renal Function and Menopausal Status with Bone Mineral Density in Middle-aged Women, 2015; A cross-sectional association between bone mineral density and parathyroid hormone and other biomarkers in community-dwelling young adults: the CARDIA study, Journal of Clinical Endocrinology & Metabolism, 2013)
- The authors state (p12, line 357) that “compared to women, men have less estrogen which plays an important role for the prevention of osteoporosis”. Can this section be expanded and can they also comment on the impact of testosterone?
Response:
Thanks for the reviewer's insightful comments. The sentence has been rephrased.
“First, compared to women, men have more androgen and less estrogen. Serum testosterone levels were found not to be associated with BMD in men[77]. Estrogen plays an important role for the prevention of osteoporosis in aged male rats[78]. Overall, serum estradiol is more related to BMD than testosterone in men[77].
- The authors also state (p12, line 366) that “, taking foods enriched with vitamin C or supplements seems 366 crucial for preventing osteoporosis in young and middle-age men”.
Please expand the discussion on this topic and whether they are any trial in this area.
Response:
Thanks for the reviewer's suggestion. Although we do not have any intervention trial in this area, our findings suggest a need for an intervention study in this area. We have included the following in “Discussion”.
The present study is a cross-sectional study that cannot determine causality between vitamin C and their effects on bone in young and early middle aged adults. Our findings suggest a need for an intervention study by taking foods enriched with vitamin C or supplements in young and middle-aged adults to assess the prevention effects on plasma vitamin C deficiency and osteoporosis.
- There are missing p-values in table 3. Can the authors please explain this and add the values if required?
Response:
Thanks for the comment. The following is our clarification.
The first p value (0.225) indicates the probability of differences among the tertile groups by ANOVA test.
The second p value (0.344) indicates the probability of the distribution differences for 20-35 years old versus 36-49 years old among three subcatogories of BMD by chi square test (2 groups X 3 subcatogories).
However, the means (SD) in the table 3 were deleted to make the table 3 easier to understand.
|
Bone mineral density of lumbar spine in men |
|
||
|
Lowest (n=297) |
Middle (n=299) |
Highest (n=297) |
P |
Age groups (years), mean (SD) |
42.2 (5.5) |
41.8 (5.4) |
42.6 (5.1) |
0.225 |
20-35, n (%) |
39 (13.1%) |
32 (10.7%) |
28 (9.4%) |
0.344 |
36-49, n (%) |
258 (86.9%) |
267 (89.3%) |
269 (90.6%) |
|
There are minor spelling errors:
- P3, line 140: “has” should be “have”
Response:
Thanks for the comment. The misspelled word is corrected as suggested.
- P9, line 291; “nutritents” should be nutrients
Response:
Thanks for the comment. The misspelled word is corrected as suggested.
Round 2
Reviewer 1 Report
Dear authors,
Thank you for your replies to my comments.
I enjoyed reading your revised manuscript.
I have no further recommendations to make.
I approve it to be published as it is.
Author Response
Dear Reviewer 1,
Thank you for the insightful suggestions.